# Sulfated CXCR3 Peptide Trap Use as a Promising Therapeutic Approach for Age-Related Macular Degeneration

**DOI:** 10.3390/biomedicines12010241

**Published:** 2024-01-22

**Authors:** Gukheui Jo, Jae-Byoung Chae, Sun-Ah Jung, Jungmook Lyu, Hyewon Chung, Joon H. Lee

**Affiliations:** 1Myung-Gok Eye Research Institute, Kim’s Eye Hospital, Konyang University College of Medicine, Seoul 07301, Republic of Korea; cooki28@kimeye.com (G.J.); ayong37@konyang.ac.kr (S.-A.J.); 2Department of Ophthalmology, Konkuk University College of Medicine, Seoul 05029, Republic of Korea; jack1690@naver.com; 3Department of Medical Science, Konyang University, Seo-gu, Daejeon 35365, Republic of Korea; lyujm5@gmail.com

**Keywords:** age-related macular degeneration, cell invasion, choroidal neovascularization, macrophage recruitment, recombinant sulfated CXCR3 peptide trap

## Abstract

Background and Objectives: Chemokines have various biological functions and potential roles in the development or progression of neuroinflammatory diseases. However, the specific pathogenic roles of chemokines in the major cause for vision loss among the elderly, the leading cause of blindness in older individuals, remain elusive. Chemokines interact with their receptors expressed in the endothelium and on leukocytes. The sulfation of tyrosine residues in chemokine receptors increases the strength of ligand–receptor interaction and modulates signaling. Therefore, in the present study, we aimed to construct a human recombinant sulfated CXCR3 peptide trap (hCXCR3-S2) and mouse recombinant sulfated CXCR3 peptide trap (mCXCR3-S2) to demonstrate in vivo effects in preventing choroidal neovascularization (CNV) and chemotaxis. Materials and Methods: We generated expression vectors for *mCXCR3-S2* and *hCXCR3-S2* with *GST* domains and their respective cDNA sequences. Following overexpression in *E. coli BL21 (DE3)*, we purified the fusion proteins from cell lysates using affinity chromatography. First, the impact of hCXCR3-S2 was validated in vitro. Subsequently, the in vivo efficacy of mCXCR3-S2 was investigated using a laser-induced CNV mouse model, a mouse model of neovascular age-related macular degeneration (AMD). Results: hCXCR3-S2 inhibited the migration and invasion of two human cancer cell lines. Intravitreal injection of mCXCR3-S2 attenuated CNV and macrophage recruitment in neovascular lesions of mouse models. These in vitro and in vivo effects were significantly stronger with CXCR3-S2 than with wild-type CXCR3 peptides. Conclusion: These findings demonstrate that the sulfated form of the CXCR3 peptide trap is a valuable tool that could be supplemented with antivascular endothelial growth factors in AMD treatment.

## 1. Introduction

Age-related macular degeneration (AMD), a condition marked by the progressive decline of the retinal pigment epithelium (RPE), retina, and choriocapillaris, stands as the leading cause of vision impairment among the elderly worldwide [1,2]. The stages of AMD are categorized into early, intermediate, or advanced. Extracellular deposits, such as drusen and subretinal drusenoid deposits, and focal hypopigmentation/hyperpigmentation are typical characteristics of early and intermediate AMD. Advanced/late AMD with choroidal neovascularization (CNV) or geographic atrophy is characterized as neovascular AMD (nAMD). As the vascular endothelial growth factor (VEGF) is a key pathogenic factor in the development of nAMD [3], intravitreal injection of anti-VEGF agents is currently the only established treatment for nAMD. However, anti-VEGF treatment has several limitations [4]. The exact pathogenic mechanism of AMD remains unclear; however, a genome-wide association study revealed that immunological disturbances, including complement abnormalities, are partially responsible for the development and progression of AMD [5]. Chemokines are chemotactic agents classified into four main subgroups: CC, CXC, CX3C, and C [6]. Chemokines have various biological functions; some perform proinflammatory activities, whereas others regulate cellular homeostasis and cell migration; accordingly, they have potential roles in the development or progression of neuroinflammatory diseases [7]. In AMD, CC chemokines, particularly CCL2, have been extensively studied. It recruits monocytes and macrophages, and elevated levels of CCL2 are associated with AMD progression [8]. Although CXC chemokines play multiple roles, including chemotaxis of immune cells as well as regulation of angiogenesis and epithelial–mesenchymal transition [9], the pathogenic roles of CXC chemokines in AMD are still not fully understood. Among the CXC chemokines, CXCR3 is a transmembrane receptor associated with G proteins. Predominantly located on immune cells, especially activated T lymphocytes and natural killer cells, CXCR3 plays a crucial role in guiding effector cells to infection sites and assisting in the clearance of pathogens, contributing significantly to immunological functions [10]. The human CXCR3 gene encodes a 45.65 kDa (415 aa) protein, and its partial cDNA was cloned from a CD4 + T lymphocyte cDNA library in 1996 [11]. In a study utilizing rapid amplification of cDNA ends (RACE) [12], Lasagni et al. identified a splice variant of human CXCR3, subsequently named CXCR3B. The original transcript was renamed CXCR3A, although it is still commonly referred to as CXCR3. Post-translational modifications of CXCR3 are crucial for its interactions with ligands. In a study employing site-directed mutagenesis [13], the impact of sulfation on CXCR3 binding and activation was identified. Specifically, the sulfation of Tyr 27 and Tyr 29 residues is necessary for CXCR3 activation, along with the crucial role of the first 16 residues of the N-terminal region for optimal binding of CXCL10 and CXCL11. CXCR3 binds to ligands such as CXCL9, CXCL10, and CXCL11 [9]. Increased levels of aqueous CXCL10 have been reported in patients with AMD [14,15]. Furthermore, elevated levels of CXCR3 and one of its ligands, CXCL10, have been observed in eyes affected by CNV compared to control eyes [14,16]. Through its interaction with the CXCR3 receptor, CXCL10 triggers chemotaxis, apoptosis, cell growth, and angiostasis. While CXCL10’s increased expression is noted in various inflammatory conditions, including infectious diseases, immune dysfunctions, and tumor development [17], its specific role in nAMD remains unclear. Considering the intimate connection between inflammation and neovascularization, we proposed that targeting CXCL10 could be a therapeutic strategy for treating CNV. Tyrosine residues in the N-terminal of chemokine receptors are post-translationally sulfated. These sulfated tyrosine residues modulate the binding to chemokine ligands that induce intracellular signaling pathways. Studies have also shown that the sulfated peptides of chemokine receptors increase the binding affinity of peptides to their cognate ligands [18,19]. In this study, we aimed to develop a sulfated CXCR3 peptide trap designed to target CXCR3 ligands like CXCL10. The goal was to demonstrate its efficacy in the murine laser-induced CNV model, specifically addressing its impact on CNV.

## 2. Materials and Methods

### 2.1. Plasmid Construction

An automated 96-well parallel-array oligonucleotide synthesizer was used to prepare oligo primers. Wild-type mouse *CXCR3* (*mCXCR3-WT*) and *mCXCR3-2S* nucleotides as well as wild-type human *CXCR3* (*hCXCR3-WT*) and *hCXCR3-2S* nucleotides were synthesized through oligo shuffling [20]. The EcoRV, BamHI, and XhoI restriction sites were introduced in the polymerase chain reaction (PCR) products, as shown within parentheses in the sequences listed in Appendix A. The synthesized double-stranded oligonucleotide was inserted into the EcoRV-, BamHI-, and XhoI-digested pET-41a vector (Novagen, Madison, WI, USA), containing an N-terminal GST, a polyhistidine (6 × His) tag, and a C-terminal polyhistidine for purification.

### 2.2. Protein Purification

The mCXCR3-WT pET-41a, mouse sulfated CXCR3 peptide trap (mCXCR3-S2) pET-41a, hCXCR3-WT pET-41a, human sulfated CXCR3 peptide trap (hCXCR3-S2) pET-41a (Appendix A), and *pSUPAR6-L3-3SY* constructs were separately transformed into *Escherichia coli BL21 (DE3)* [21]. The *mCXCR3-WT* and *hCXCR3-WT* transformants were induced with 0.1 mM isopropyl β-d-1-thiogalactopyranoside (IPTG, final concentration) at 37 °C for 5 h. The *mCXCR3-S2* and *hCXCR3-S2* transformants were inoculated in 250 mL of Luria–Bertani medium containing 10 mM sulfotyrosine (Bachem, Bubendorf, Switzerland) and cultured until OD_600_ = 1. For the overexpression of the fusion proteins, 1 mM IPTG was added and cultures were incubated for 20 h at 25 °C. After centrifugation of the cultures at 10,000× *g* for 10 min at 4 °C, the bacterial pellets were resuspended in 10 mL of binding buffer (5 mM imidazole, 0.5 M NaCl, and 20 mM Tris-HCl; pH 7.9) and sonicated. The supernatant, containing the target material, was applied to a pre-equilibrated Ni-NTA resin (Qiagen, Hilden, Germany) using distilled water and binding buffer, allowing gravity to facilitate flow. The column underwent three washes with 10 volumes of binding and elution buffer (100 mM imidazole, 0.5 M NaCl, and 20 mM Tris-HCl; pH 7.9) to remove any residual imidazole before dialysis (20 mM Tris-HCl, 50 mM NaCl, and 0.5 mM β-mer; pH 7.5) for 1 h. Subsequently, the dialyzed fusion proteins were separated using 10% sodium dodecyl sulfate–polyacrylamide gel electrophoresis (SDS–PAGE). The gels were stained with Coomassie Blue R-250 and destained with 30% methanol and 10% acetic acid in distilled water.

### 2.3. Wound Healing Assay

SKOV3 human ovarian and MDM-MB-231 human breast cancer cell lines were obtained from the Korean Cell Line Bank (Seoul, Republic of Korea) and seeded in 24-well plates (SKOV3, 2 × 10⁵ cells/well; MDM-MB-231, 3 × 10⁵ cells/well). When the cells reached 100% confluence, cell monolayers were vertically wounded by scratching with a 200 μL sterile pipette tip. To prevent any impact on the cell growth rate, the cell culture medium transitioned from RPMI-1640 supplemented with 10% fetal bovine serum to serum-free RPMI-1640 (Invitrogen, Carlsbad, CA, USA). The cells were then incubated with or without 100 ng/mL CXCL10 (R&D Systems, Minneapolis, MN, USA) and 10 μg/mL hCXCR3 WT or hCXCR3-S2. Phase-contrast images were taken both immediately after scratching and following 24 h of incubation at 37 °C using a DP70 microscope (Olympus, Shinjuku, Japan).

### 2.4. Invasion Assay

The invasion assay employed an 8 μm-pore 24-well Transwell plate (Sigma Aldrich, St. Louis, MO, USA), coated with phosphate-buffered saline (PBS) and containing 25 μg Matrigel (Sigma Aldrich) and 0.1% gelatin (Sigma Aldrich). Cells, reaching 80% confluence in a growth medium, were synchronized through a 24 h starvation in serum-free RPMI-1640 medium. Subsequently, they were seeded (2 × 10⁵ cells/mL) in the top chamber [22]. The bottom chamber contained serum-free medium with or without 200 ng/mL CXCL10 and with or without 10 μg/mL hCXCR3 WT or hCXCR3-S2. Following incubation, cells were fixed in 100% methanol for 10 min and stained with 0.1% crystal violet (Thermo Fisher Scientific, Waltham, MA, USA) for an additional 10 min. Cells remaining on top of the filter were removed using a cotton swab, while those that had migrated to the underside of the filter were micrographed using a DP70 microscope (Olympus). Three fields per sample were captured at 10× magnification. 

### 2.5. RAW 264.7 Cell Culture

RAW 264.7 cells, a murine macrophage cell line, were purchased from the Korean Cell Line Bank (Republic of Korea). The cells were cultured in DMEM (Welgene, Gyeongsan, Republic of Korea) supplemented with 10% fetal bovine serum (FBS) (Gibco, Billings, MT, USA) and 1% penicillin/streptomycin (Welgene). Cultures were maintained in a humidified incubator at 37 °C with 5% CO_2_. To stimulate an inflammatory response, 1 × 10^6^ RAW 264.7 cells were plated in a 60 mm dish and exposed to 1 μg/mL of LPS (Sigma) for a duration of 24 h.

### 2.6. Experimental Animals

Eight-week-old male C57BL/6J mice were purchased from Charles River Laboratories Japan (Yokohama, Japan). All mice were housed in microisolator cages under specific pathogen-free conditions and maintained under a 12-hour light/12-hour dark cycle in a humidity- and temperature-controlled facility with ad libitum access to food and water.

### 2.7. Laser-Induced CNV Mouse Models

C57BL/6J mice underwent anesthesia using a 4:1 mixture of Zoletil (Virbac, Carros Cedex, France) and xylazine (Bayer Healthcare, Leverkusen, Germany) with added topical 0.5% proparacaine (Alcaine; Alcon, Geneva, Switzerland) for topical anesthesia. Pupils were dilated by Tropherine eye drops (0.5% tropicamide and 0.5% phenylephrine hydrochloric acid; Hanmi Pharm, Seoul, Republic of Korea). A slit-lamp delivery system (SL-1800; NIDEK, Tokyo, Japan) with a green laser photocoagulator (GYC-500; NIDEK) (532 nm laser, 50 μm spot size, 0.1 s duration, 200 mW) was used to generate three laser spots in each eye using 12 mm-diameter microscope cover glasses (Paul Marienfeld GmbH, Lauda-Königshofen, Germany) as contact lenses while protecting the optic nerve with a lubricant (hypromellose; SAMIL, Seoul, Republic of Korea). Successful Bruch’s membrane disruption was indicated by gaseous bubbles. After photocoagulation, intravitreal injections of GST, mCXCR3-WT (2 µg/µL), mCXCR3-S2 (2 µg/µL), and aflibercept (EYLEA, 2 µg/µL; Bayer Healthcare) were administrated. Only spots with accompanying bubbles were included in the study. 

### 2.8. Quantitative Reverse Transcriptase-Mediated Real-Time PCR (qPCR)

Total RNA from RAW 264.7 cells and the RPE/choroid was extracted using Trizol reagent (Thermo Fisher Scientific, Waltham, MA, USA). The mRNA from the isolated RNA samples was then reverse-transcribed into cDNA using iScript reverse transcriptase (Bio-Rad Laboratories, CA, USA). Quantification of mRNA expression was performed using SsoAdvanced Universal SYBR Green Supermix (Bio-Rad) on a CFX96 Real-Time PCR detection system (Bio-Rad). Each sample group underwent Real-Time PCR in triplicate. Details on the primers utilized for PCR are provided in Appendix A.

### 2.9. Quantitation of CNV in the Mouse Model of Laser-Induced CNV

FITC-dextran (Sigma Aldrich) was intravenously injected into mice for fluorescent labeling. After 3 min, the mice were euthanized with CO_2_ gas, and their eyes were extracted. The anterior parts of the eye and the neural retina were quickly separated from the RPE/choroid. The RPE/choroid was then fixed in 4% paraformaldehyde for 30 min, rinsed three times with PBS, and cut radially for mounting (Polysciences, Inc., Warrington, PA, USA). These prepared tissues were positioned on slides for imaging via a confocal microscope (FluoView 1000; Olympus). The collected Z-section images were further processed and analyzed using specialized software (Metamorph version 7.7.3.0; Molecular Devices, Sunnyvale, CA, USA).

### 2.10. Immunofluorescence Staining

The RPE/choroid tissues were fixed with 4% paraformaldehyde for 30 min at 25 °C and permeabilized with 0.1% Triton X-100 in PBS for 10 min. After blocking with 1% bovine serum albumin in PBS for 1 h, the fixed tissues were incubated overnight at 4 °C with primary antibodies against F4/80 (1:200; eBioscience, San Diego, CA, USA). The stained tissues were washed with PBS for 5 min and incubated for 2 h at a room temperature. with Alexa Fluor 555-conjugated goat anti-mouse IgG (1:1000; Thermo Fisher Scientific) secondary antibody. After incubation with the secondary antibody, the tissues were stained with the nuclear dye 4′,6-diamidino-2-phenylindole (1:3000; Thermo Fisher Scientific) in PBS for 10 min at room temperature. The tissues were then mounted using a mounting medium (Agilent Dako, Danvers, MA, USA). The stained tissues were observed under a super-resolution confocal laser scanning microscope (LSM 800; Carl Zeiss, Oberkochen, Germany) and the stained cells were observed under an inverted microscope (#DMi1; Leica Microsystems, Wetzlar, Germany). 

### 2.11. Statistical Analysis

All experimental data are presented as mean ± standard deviation. Statistical significance (*p*-value) was determined using an unpaired two-tailed Student’s *t*-test followed by Fisher’s least significant difference post hoc test or Tukey’s multiple comparisons test. Data were analyzed using GraphPad Prism 5 software version 5.01 (GraphPad, San Diego, CA, USA). Statistical significance was set at *p* < 0.05 (* *p* < 0.05, ** *p* < 0.01, and *** *p* < 0.001).

## 3. Results

### 3.1. Cloning and Purification of mCXCR3-S2 and hCXCR3-S2

To produce mCXCR3-S2 and hCXCR3-S2 proteins, we PCR-amplified the *mCXCR3-S2* and *hCXCR3-S2* using oligo shuffling. To overexpress and purify the mCXCR3-S2 and hCXCR3-S2 proteins, we constructed *mCXCR3-S2* and *hCXCR3-S2* expression vectors containing the *GST* domain and respective cDNA sequences. In addition, we constructed *mCXCR3-WT* and *hCXCR3-WT* expression vectors containing the GST protein domain and respective cDNA sequences (Figure 1a). We used the *pSUPAR6-L3-3SY* plasmid to express both *mCXCR3-S2* and *hCXCR3-S2* (Figure 1c). Each recombinant protein contained a six-histidine residue tag. After overexpressing the fusion proteins in *E. coli BL21 (DE3)*, we purified the mCXCR3-S2 and hCXCR3-S2 proteins from the cell lysates using affinity chromatography and a metal-chelating matrix under urea-denaturing conditions. Both mCXCR3-S2 and hCXCR3-S2 were observed as single bands on SDS-PAGE. Notably, mCXCR3-S2 and hCXCR3-S2 could be distinguished from mCXCR3-WT and hCXCR3-WT by their reduced migration caused by the additional presence of sulfur moieties (Figure 1b) [21].

### 3.2. hCXCR3 Sulfation Attenuated CXCL10-Induced Cell Migration and Invasion

CXCR3 is highly expressed in metastatic cancer cells, and its binding with chemokine ligands, such as CXCL10, enhances the migration and invasive motility of cells [23,24,25]. To test the inhibitory effect of recombinant CXCR3-S2 and WT peptides on the interaction between CXCR3 and chemokines, we determined whether hCXCR3-S2 affected the CXCL10-induced migration and invasion of highly metastatic SKOV3 and MDA-MB-231 cells using a scratch wound healing assay. After creating a scratch wound in the cell monolayer, the cells were treated with CXCL10, CXCL10, and hCXCR3-WT, or hCXCR3-S2. CXCL10 stimulation increased cell migration to the wound area after 24 h. However, treatment with hCXCR3-S2 inhibited the CXCL10-induced migration, suggesting that hCXCR3 sulfation suppressed the CXCL10-induced migration of SKOV3 and MDA-MB-231 cells (Figure 2a,b). We also performed Transwell invasion assays to evaluate the effect of hCXCR3 sulfation on the invasive abilities of SKOV3 and MDA-MB-231 cells. We observed that medium containing CXCL10 in the bottom of the Transwell chamber efficiently induced the invasion of SKOV3 and MDA-MB-231 cells from the upper chamber. However, when hCXCR3-S2-containing medium was present in the bottom chamber, the CXCL10-induced invasion of SKOV3 and MDA-MB-231 cells was inhibited, indicating that hCXCR3 sulfation suppressed the CXCL10-induced motility of cells (Figure 2c,d).

**Figure 1 biomedicines-12-00241-f001:**
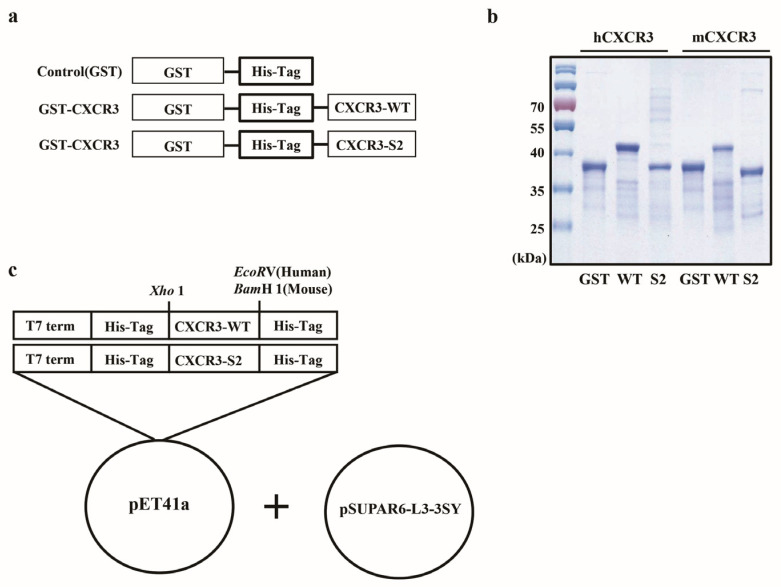
Cloning and purification of CXCR3 fusion proteins in *Escherichia coli*. Diagram of the control, CXCR3-WT, and CXCR3-S2 fusion proteins. CXCR3 was cloned into the EcoRV (human) or BamHI (mouse) and XhoI sites of *pET41a.* The expression vector was controlled by the T7 promoter. Expression was induced by the addition of IPTG (**a**). Purified fusion proteins were detected using Coomassie brilliant blue staining on 10% sodium dodecyl sulfate–polyacrylamide gels (**b**). Map of the optimized *pSUPAR6-L3-3SY* plasmid encoding the components necessary for the translational incorporation of sulfotyrosine in response to the TAG codon (**c**).

### 3.3. mCXCR3-S2 Prevented CNV and Macrophage Infiltration in the Laser-Induced CNV Mouse Model

To demonstrate the in vivo efficacy of the mCXCR3-S2 trap peptide in inhibiting CNV formation, we first assessed the expression of CXCR3 and its ligands in a laser-induced CNV mouse model. The mRNA expression levels of CXCR3 and its ligands, including CXCL10, were elevated in the RPE/choroid of the CNV model compared to the control group (Appendix A). We then administered intravitreal injections of GST, mCXCR3-WT, mCXCR3-S2, and EYLEA into mice with laser-induced CNV. Seven days post-laser treatment, 3D analysis of FITC-dextran-labeled flat-mounted RPE/choroid tissues revealed a significant reduction in CNV volume in the mCXCR3-S2-treated group compared to the GST- and mCXCR3-WT-treated groups (Figure 3). The upregulation of CXCR3 ligands in CNV mice may contribute to the recruitment of immune cells, including macrophages, to the site of the pathological CNV lesions. Moreover, previous research has emphasized the interplay between CXCR3 and macrophages, highlighting the receptor’s critical role in macrophage-driven inflammation [26,27,28]. Consequently, we examined macrophage-related gene expression in the laser-induced CNV model and in RAW 264.7 mouse macrophage cells stimulated with LPS. We observed an upregulation of macrophage markers, including F4/80, MERTK, and CD11b, in the RPE/choroid of the CNV model compared to controls. In LPS-stimulated RAW 264.7 cells, there was a concomitant increase in the expression of CXCR3, its ligands, and macrophage-associated genes (Appendix A). Additionally, we evaluated macrophage infiltration into the laser injury site. Immunofluorescence staining for F4/80 indicated a significant decrease in F4/80-positive cells within the fibrovascular CNV lesions of eyes treated with mCXCR3-S2 compared to those treated with GST or mCXCR3-WT (Figure 4). These results suggest that the mCXCR3-S2-mediated inhibition of macrophage infiltration may correlate with the observed reduction in fibrovascular formation.

## 4. Discussion

In this study, we explored the impact of the tyrosine-sulfated CXCR3-S2 peptide trap on malignant tumor cell behavior in comparison to the wild-type CXCR3 peptide trap, using MDA-MB-231 and SKOV3 cells. Through a wound healing assay and Transwell cell invasion assay, we demonstrated that the sulfated CXCR3-S2 peptide trap significantly reduced CXCL10-induced cancer cell migration and invasion (Figure 1 and Figure 2). CXCR3 has been identified in numerous malignant cell lines and is linked to the prognosis of individuals with melanoma, colon cancer, and breast cancers [29,30,31]. Notably, elevated CXCR3 expression in melanoma, colon, and breast cancers is indicative of more malignant and aggressive tumor characteristics. Various studies have highlighted that tyrosine sulfation of different chemokine receptors enhances the binding affinity to their respective chemokines [19,32,33,34,35]. For instance, the sulfation of tyrosine residues on the N-terminus of CXCR3 has been shown to modulate the interaction between CXCL9, CXCL10, CXCL11, and CXCR3, activating CXCL9, CXCL10, CXCL11-CXCR3 signaling [13]. Additionally, Gao et al. demonstrated that sulfation of tyrosine 27 and 29 in the N-terminal region of human CXCR3 increases the binding affinity for CXCL10, indicating that CXCR3 sulfation plays a crucial role in the interaction of CXCL10-CXCR3 [36]. These findings are consistent with our results, indicating that sulfation of the CXCR3 peptide trap reduces CXCL10-induced cancer cell migration and invasion by binding to its ligand, CXCL10. 

Intravitreal administration of the fusion proteins significantly reduced CNV volume in a laser-induced CNV mouse model compared to wild-type CXCR3 and GST control, further confirming the significant role of the CXCR3 signaling axis in CNV formation (Figure 3, Appendix A). Previously, Fujimura et al. reported increased CXCR3 and IP-10 mRNA in laser-induced models, noting larger CNV in CXCR3-deficient mice and worsening CNV with anti-CXCR3/anti-IP-10 antibodies [16], suggesting a protective role for CXCR3 in CNV. This contrasts with our findings, where recombinant CXCR3-S2 peptide traps significantly reduced CNV and macrophage recruitment compared to wild-type CXCR3 proteins. Discrepancies may arise from several factors: increased macrophage recruitment and elevated CCL2 levels in CXCR3-deficient mice potentially leading to larger CNV, minimal angiostatic effects of recombinant IP-10 on CNV in wild-type mice, and the possibility that anti-CXCR3/anti-IP-10 antibodies used were not fully validated as neutralizing agents, thereby affecting interpretation. While our study demonstrated the efficacy of recombinant CXCR3-S2 peptide traps in reducing CNV and macrophage recruitment, Denoyer et al. showed that a small molecule CXCR3 antagonist in a rat model of ocular hypertension decreases intraocular pressure, prevents retinal neurodegeneration, and preserves visual function by restoring trabecular function [37]. While it remains debatable, given the membrane structure of CXCR3—a G protein-coupled receptor with transmembrane domains—a CXCR3 trap peptide might be a more effective strategy compared to small molecule inhibitors.

Importantly, the mCXCR3-S2 peptide trap specifically targets CXCR3 receptor ligands on cells. This trap peptide directly binds to CXCR3 ligands like CXCL10 found near inflammation sites, mimicking the CXCR3 receptor binding site. Consequently, the CXCR3-S2 peptide trap operates by eliminating ligands for the CXCR3 receptor on cells, thereby preventing the recruitment of inflammatory cells, including macrophages. These results were consistent with those of previous studies, which reported that the sulfation of chemokine receptors plays a critical role in their signaling [13,38,39]. Several research findings indicate that the recruitment of macrophages plays a significant role in the development of CNV in AMD [1,40,41]. In our study, diminished recruitment of F4/80^+^ cells was observed in sulfated CXCR3 antibody-injected mice. Whether macrophages play a protective or aggravating role in CNV remains unclear; however, macrophages trigger VEGF secretion in the angiogenic environment of the choroid [42,43]. We speculated that a sulfated CXCR3 peptide could capture CXCL10 before the latter could induce the chemotaxis of M1 macrophages [44] and could therefore exert a protective effect against CNV. It should be noted that the incorporation of sulfotyrosine, an unnatural amino acid, has some drawbacks, such as low yield, low permeability, and leaky suppression [15]. Further studies are warranted to overcome these limitations. 

## 5. Conclusions

Intravitreal application of recombinant CXCR3-S2 peptide traps reduced CNV and macrophage recruitment in a mouse CNV model, indicating CXCL10-CXCR3 axis targeting as a potential nAMD therapy. Further studies on CXCR3-S2 peptide traps for AMD and CXCR3-related cancers are encouraged to explore broader therapeutic applications.

## Figures and Tables

**Figure 2 biomedicines-12-00241-f002:**
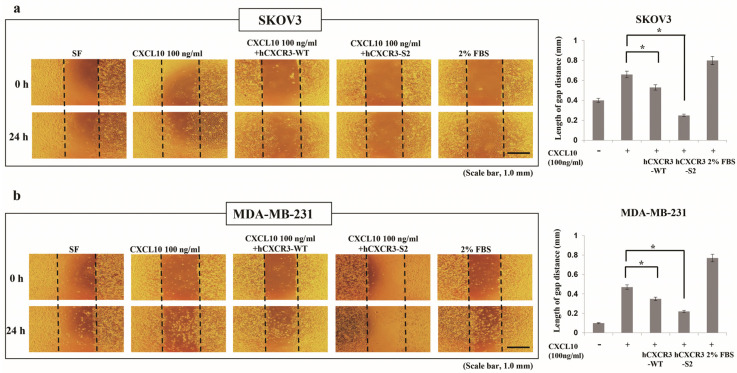
hCXCR3-S2 inhibits the CXCL10-induced migration and invasion of SKOV3 and MDA-MB-231 cells. A scratch wound healing assay was performed using SKOV3 (2 × 10⁵ cells/well) (**a**) and MDA-MB-231 (3 × 10⁵ cells/well) cells (**b**) treated with or without 100 ng/mL CXCL10 and 10 μg/mL hCXCR3-WT or hCXCR3-S2 to determine their migration ability. CXCL10, hCXCR3-WT, or hCXCR3-S2 was added immediately after wound creation. The cells were allowed to migrate for 24 h, and photographs were taken before and after migration (magnification, ×4). Compared to hCXCR3-WT, hCXCR3-S2 significantly and more effectively inhibited the CXCL10-induced migration of the two cell lines. Scale bar, 1.0 mm. CXCL10, hCXCR3-WT, or hCXCR3-S2 was added immediately after wound creation. The average distance between the edges of the wound was measured in three independent experiments. hCXCR3-S2 significantly and more effectively inhibited the CXCL10-induced migration compared with hCXCR3-WT. The results are presented as mean ± standard deviation of three independent experiments (* *p* < 0.05). Invasion assays were performed using SKOV3 (**c**) and MDA-MB-231 (**d**) cells in Transwell chambers with Matrigel-coated membranes. The cells were loaded in the upper well of the chamber, and 200 ng/mL CXCL10 and 10 μg/mL hCXCR3-WT or hCXCR3-S2 were added to the lower well of the chamber. After 24 h, the cells that migrated to the lower well were stained with crystal violet and photographed (magnification, ×10). Compared to hCXCR3-WT, hCXCR3-S2 significantly and more effectively inhibited the CXCL10-induced invasion of these cells. Scale bar, 1.0 mm. The statistical graph presents the density of invaded cells per field, 24 h after seeding. hCXCR3-S2 significantly and more effectively inhibited the CXCL10-induced cell invasion compared with hCXCR3-WT. The results are presented as mean **±** standard deviation of three independent experiments (* *p* < 0.05).

**Figure 3 biomedicines-12-00241-f003:**
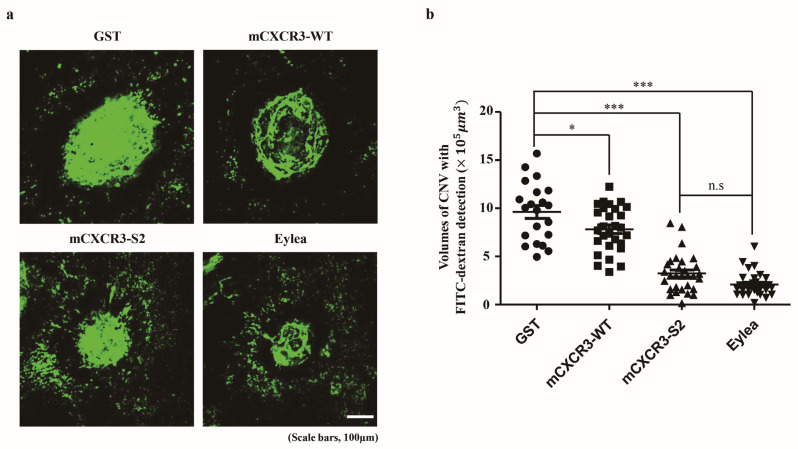
Effect of mCXCR3-S2 in the laser-induced choroidal neovascularization (CNV) mouse model. Representative fluorescent images of CNV are shown in FITC-dextran-labeled RPE/choroidal flat mounts from mice intravitreally injected with GST (2 µg/µL), mCXCR3-WT (2 µg/µL), mCXCR3-S2 (2 µg/µL), or EYLEA (aflibercept, 2 µg/µL) at 7 d after laser application. Each of these groups received an injection volume of 1 µL. The number of mice per group was 5, and the results obtained from a total of 10 eyes were analyzed (**a**). Quantification of CNV volume. (Laser spots with choroidal hemorrhage or combined with each other were excluded from the quantification. Number of laser spots in WT = 21, number of laser spots in GST = 28, number of laser spots in S2 = 30, and number of laser spots in EYLEA = 29; *** *p* < 0.001, * *p* < 0.05) Scale bar = 100 µm (**b**).

**Figure 4 biomedicines-12-00241-f004:**
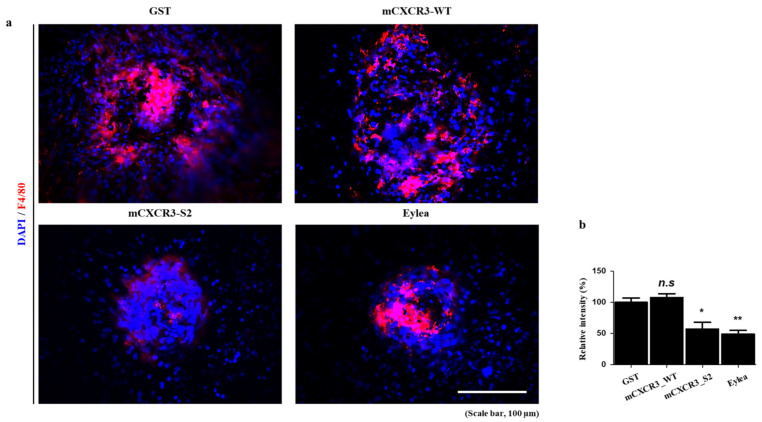
Immunofluorescence of F4/80 in the laser-induced CNV mouse model. Representative fluorescent images of RPE/choroidal flat mounts from mice intravitreally injected with GST, mCXCR3-WT, mCXCR3-S2, or EYLEA (aflibercept) at 7 d after laser application. Mouse RPE/choroidal flat mounts were stained with 4′,6-diamidino-2-phenylindole (blue) and F4/80 (red) antibodies (*n* = 4 per group) (**a**). Quantification of F4/80 fluorescence intensity. (** *p* < 0.01, * *p* < 0.05) (**b**). Scale bar = 100 µm.

## Data Availability

Data are contained within the article or Appendix A.

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
