# Peer review of "Sulfated CXCR3 Peptide Trap Use as a Promising Therapeutic Approach for Age-Related Macular Degeneration"

_biomedicines, 2024, doi:10.3390/biomedicines12010241_

Round 1
Reviewer 1 Report (Previous Reviewer 2)
Comments and Suggestions for Authors
The authors are answered all the questions.
Author Response
.
Reviewer 2 Report (New Reviewer)
Comments and Suggestions for Authors
At present manuscript submitted by Lee et al., is highlighted on sulfated CXCR3 peptide trap as a promising therapeutic approach for age-related macular degeneration. It is very interesting for the age-related macular degeneration which has not enough therapeutic strategy in clinic. However, the following question should be conceived.
1. The abstract should be supplemented in materials and methods, in fact, the constructed CXCR3 should be introduced in the presented abstract.
2. Introduction: there is not enough data or information for the scientific hypothesis. Please supplemented the data or information in logic.
3. The experimental designed should be completed.
(1) The authors should be supplemented the data of CXCR3 identification. At present, the authors just provided the coomassie brilliant blue staining on 10% SDS-PAGE. Please provided the MS data, etc.
(2) The wound healing assay and invasion assay should be presented the picture.
(3) Please supplemented the pharmacodynamics of the animal model. At present, the data is too little to confirm the pharmacological effect.
4. The discussion should be widely revised with the results.
Comments on the Quality of English LanguageThe language should be improved.
Author Response
.

This manuscript is a resubmission of an earlier submission. The following is a list of the peer review reports and author responses from that submission.
Round 1
Reviewer 1 Report
Comments and Suggestions for Authors
In this paper, the authors construct recombinant sulfated CXCR3 peptide trap and investigated the effects in two cancer cell lines and a mouse choroidal neovascularization model. They found that intravitreal injection of mCXCR3-S2 attenuated CNV formation and macrophage recruitment. Overall, the results are interesting. However, there are several issues that should be addressed.
1. CNV formation is mainly included choroidal endothelial cells and local surrounding cells. The authors used SKOV3 human ovarian and MDM-MB-231 human breast cancer cell lines to test the effect of CXCR3-S2 on cell migration. These could not effectively represent its effect on vascular endothelial cells. The authors should use HUVEC or choroidal endothelial cells to see if CXCR3-S2 could alter the cell proliferation, migration, and tube-formation.
2. In Figure 3, the authors found intravitreal injection of mCXCR3-S2 attenuated CNV formation. Then which cell did mCXCR3-S2 mainly bind to or interact with to exert the inhibitory effects?
3. In Figure 4, the authors showed the number of F4/80-labeled cells was reduced in CNV lesions after intravitreal injection of mCXCR3-S2. Did this result indicate the mCXCR3-S2 attenuated CNV formation through inhibition of macrophage recruitment? The authors should test the effect of CXCR3-S2 on macrophage activation and factor secretion in vitro, and design macrophages stimulation/inhibition experiment in vivo to further validated the results.
Reviewer 2 Report
Comments and Suggestions for Authors
The manuscript “Sulfated CXCR3 peptide trap use as a promising therapeutic approach for age-related macular degeneration” by G. Jo et al. describes the production of a sulfated CXCR3 peptide trap and its evaluation in cells and in age-related macular degeneration (AMD) model in vivo in mice.
The manuscript is clear and well written, and the conclusions are supported by the data. Suitable control experiments including a non-sulfated CXCR3 trap are provided. This work will deserve publication after the following corrections.
- In the introduction, the authors should give some more information regarding CXCR3: for example, what is known about its structure, splice variants or isoforms, size, localization, levels of expression? A useful reference could be “Satarkar D and Patra C (2022) Evolution, Expression and Functional Analysis of CXCR3 in Neuronal and Cardiovascular Diseases: A Narrative Review. Front. Cell Dev. Biol. 10:882017. doi: 10.3389/fcell.2022.882017”. The full name CXC chemokine receptor 3 (CXCR3) should be given.
- A figure could be added in the introduction to show these aspects in a schematic cell, displaying the CXC chemokines ligands, the CXCR3 (sulfated or not sulfated), and the strategy developed in this study. The sulfated CXCR3 peptide trap is expected to trigger an anti-angiogenic response, preventing CNV.
- For the protein purification in 2.2., is there a folding step? Does the CXCR3 peptide traps, sulfated or not, require folding into a tertiary structure? Are there disulfide bonds in the native receptor and in the receptor expressed in bacteria? The size, the sequence, and at least the number of amino acids should be given. CXCR3 is a transmembrane G protein-coupled receptor with 7 transmembrane domains and intra- and extra- cellular domains. The tertiary structure of this receptor as expressed by the authors could be very different from its structure in the cell membrane.
- In 2.6 the volumes of injections must be given. Are the injected volumes identical for all experiments? Moreover, the molecular weights of the CXCR3 peptide traps must be given. It will allow the calculation of the concentration of injected products, to compare the efficiency of the peptide traps with aflibercept for example (molecular weight 115 kDa).
- I do not understand what is the “CXCR3-GST” used as a control. Ex. line 135, line 230, in the figure 3b and in figure 4. It seems that all these should be replaced with GST only, or PBS buffer? In figure 3a the control is PBS buffer but in the 3b it is mCXCR3-GST?
- Line 204 I do not understand if the hCXCR3-S2 induced the invasion, as written in the text. According to the figure and the discussion, it should not. In addition, there is apparently a mistake line 204: CXCR2-WT and CXCR2-S2 instead of CXCR3-WT and CXCR3-S2. Please check carefully.
Round 2
Reviewer 1 Report
Comments and Suggestions for Authors
In their revised manuscript, the authors responded to some comments provided in the original review’s. They included more discussions for helping explain the results. However, there are still some issues that should be addressed.
1. The authors mentioned mCXCR3-S2 trap peptide binds to CXCR3 ligands to inhibit choroidal neovascularization formation. Then did the authors observed whether the CXCR3 ligands increased in laser induced CNV?
2. If the authors try to comfirm mCXCR3-S2 reduces CNV formation by inhibition of macrophage recruitment, then macrophage activation experiment in vitro and stimulation/inhibition experiment in vivo is necessary.
3. The study is based on laser induced mouse CNV model, which has much difference with human age-related macular degeneration. All conclusion should limited to laser induced CNV, and can not extend to AMD treatment so far.